# Impacts on Students' Academic Performance Due to Emergency Transition to Remote Teaching during the COVID-19 Pandemic: A Financial Engineering Course Case Study

**Rezvan Nazempour** [1] , **Houshang Darabi** [1,*] **and Peter C. Nelson** [2]

1 Mechanical and Industrial Engineering, University of Illinois at Chicago, Chicago, IL 60609, USA; rnazem2@uic.edu
2 Department of Computer Science, University of Illinois at Chicago, Chicago, IL 60609, USA; nelson@uic.edu
* Correspondence: hdarabi@uic.edu

**Abstract:** The COVID-19 pandemic has enforced higher education institutions to adopt emergency remote teaching (ERT) as the substitution for traditional face-to-face (F2F) classes. A lot of concerns have been raised among education institutions, faculty, and students regarding the effectiveness of this sudden shift to online learning. This study aims to statistically investigate the impacts of such a transition on the academic performance of undergraduate students enrolled in the Financial Engineering course. A novel rank percentage measure is proposed and employed to compare the academic performance of around 500 students who attended the course during the four semesters, including the transitional disrupted semester by the pandemic, two consecutive online semesters, and the traditional face-to-face classroom. Our analysis emphasizes the significance of the differences between specific subgroups of the students. In particular, academically average to good students with cumulative GPAs greater than 2.90 have been negatively impacted by the transition to online learning, whereas the results for students with cumulative GPAs less than 2.90 are not very conclusive. Realizing the effects of such closures on the academic performance of students is considered important, since the results might have some merits for other courses and instructors. The template model can be transferred to other courses, and employed by the university administrators, specifically for developing policies in emergency circumstances that are not limited to pandemics.

**Keywords:** COVID-19; emergency remote teaching; face-to-face classes; academic performance

## 1. Introduction

The novel coronavirus disease (COVID-19) has spread globally and has affected various aspects of daily human life routines. To control the transmission of the infection, and flatten the curves, strategies such as staying at home and lockdowns have been employed. On 11 March 2020, and after the World Health Organization (WHO, Geneva, Switzerland) declaration of the pandemic [1], higher education institutions in the United States began to close in-person classes. State-wide stay-at-home orders were also designed to slow the spread of the virus. The education system, as one of the most crucial parts of society, has seen considerable disruption by the outbreak [2,3]. According to the United Nations Educational Scientific and Cultural Organization (UNESCO, Paris, France) statement, over 6 billion learners across more than 190 countries were seriously affected, in terms of education, during the peak of the crisis [4]. In addition, it highlighted that 24 million students are at risk of dropping out. In the United States, at least 14 million students have been affected by the closure of more than a thousand colleges and universities by 26 March 2020 [5].

In these circumstances, the transition to online platforms and distance learning seems to be the only feasible and attractive alternative. Therefore, higher education institutions have been enforced to employ e-learning as the substitution for traditional face-to-face

(F2F) classes [6,7]. Despite investments in learning technologies and online learning management systems, universities suffered from the loss of contingency plans appropriate to the emergency transition caused by the pandemic. Faculties and students worldwide were pushed to swiftly adopt remote education using synchronous and/or asynchronous online classes. The transition to remote teaching was stressful, since neither faculty nor students were completely prepared for this quick change, and the shift heavily relied on the ability to access or use online learning and teaching tools. Besides, some institutions lacked faculty with online teaching experience [8].

Online learning is defined by a majority of researchers as access to learning experiences using some sort of technology [9]. It is a learning process that provides learners agency, responsibility, flexibility, and choice, and to develop an effective learning ecology, careful planning, designing, and determination of goals are required [10]. However, educational experts have argued that the transition to digital settings resulting from the COVID-19 pandemic cannot be considered as "online learning". Therefore, a new concept of "emergency remote teaching" (ERT) has been defined [11,12], which is a temporary solution to an immediate problem. Such distinction plays an important role in the prosperity of distance education in a post-COVID world [10].

Synchronous and asynchronous are two types of online instruction modes when considering synchrony [13]. Blended learning (BL), on the other hand, refers to combining onsite and online learning to provide flexibility to learners, instructors, and educational institutions [11].

The impact of different teaching modalities, including face-to-face (F2F), blended (BL), and online learning, on students' academic performance has received considerable attention in educational research for decades. The literature shows that the results depend on the type of analysis, study samples such as single or multiple courses, and graduate or undergraduate level of courses [13–22]. For example, Ladyshewsky's findings and Cavanaugh et al.'s analysis over 9 and 5000 courses, respectively, confirm that by increasing the number of analyzed courses, students achieved better grades in online learning compared to those in F2F classes [21,22]. In addition, Skylar investigated the impact of synchronous and asynchronous environments on student achievement and satisfaction. The results suggest that both types of instructions are effective; however, the majority of students would prefer synchronous lectures instead of asynchronous ones [23].

During the ERT caused by the COVID-19, Chaka conducted a study to review how selected higher education institutions in the U.S. and South Africa switched to online learning, and which online tools and resources they used [24]. The findings revealed that mainly two types of online tools and resources have been employed by a majority of institutions: video conferencing platforms and learning management systems (LMS). Zoom, Canvas, Blackboard (Collaborate), Panopto, and Microsoft Teams were considered the most used online tools by U.S. universities. In addition, Blackboard (Collaborate), institutional LMSes, WhatsApp, Zoom, and Moodle were the most embraced online tools employed by South African universities [24].

In the review of emergency remote teaching due to the COVID-19 pandemic, Mishra et al. critically analyzed the publications using a range of scientometric techniques. They reported that quantitative methods were the most popular research methodology used by the researchers (43.6%), followed by qualitative (13.33%), and mixed methods (9.09%). However, the research methodology was not indicated by a large proportion of publications (33%) [25]. Khansal et al. conducted a scoping review on organizational adaptation during the early stages of the pandemic [26]. The study highlights that due to maintaining educational activities during the pandemic, instructors actively employed various methods and strategies. A survey conducted by Dios and Charlo regarding students' perceptions and opinions of F2F and e-learning caused by COVID-19 reveals that students prefer to continue with F2F learning instructions rather than online teaching or BL [27]. Aristovnik et al. presented a large-scale study on the impacts of the first wave of COVID-19 pandemic on the life of a sample of 30,383 students from 62 countries using an online questionnaire [28].

The study reveals students' satisfaction and perception of various aspects/elements of their lives during the pandemic, such as their opinions on the immediate and distant future [28].

Regarding the impact of emergency remote teaching due to the COVID-19 pandemic on students' academic performance, it seems there is no conclusive agreement in the literature. Engelhardt et al. compared the performance of students in the disrupted semester by COVID-19 to that of three previous unaffected semesters [29]. They concluded that there were no significant differences in students' performance throughout the semesters. They identified not only no measurable impact for the low-income, first-generation, and minority students, but also women overperformed in the disrupted semester compared to previous terms. Alam and Asimiran conducted an evidence-based study to compare academic and job-readiness of graduates using an empirical survey with a sample of 240 people (before and during COVID-19) [30–32]. The findings reveal that better academic scores were achieved by during-pandemic students compared to pre-pandemic ones, whereas pre-pandemic counterparts performed better in terms of job-readiness [30]. Moreover, a study conducted by Iglesias-Pradas et al. shows an increase in students' academic performance in ERT [33]. The analysis supports the idea that successful ERT implementation may be contributed to the organizational factors.

In this research study, we investigate the impact of the pandemic mid-semester disruption on the academic performance of students attending a Financial Engineering course. The Financial Engineering course (IE201) at the University of Illinois at Chicago is one of the important undergraduate courses in the College of Engineering, which is taken by four different majors, including Industrial Engineering, Mechanical Engineering, Civil Engineering, and Engineering Management. A sample data set of around 500 students is employed to conduct the analysis. The students attended the course in a transitional disrupted semester by the pandemic, two consecutive online semesters, and a traditional face-to-face semester. The course was taught by the same instructor in all semesters. This study does not represent a generic model to compare all teaching modalities for all courses. Moreover, we do not aim to develop a general approach for comparing in-person, blended, and online instructional modalities. The purpose of this study is to report insightful analysis, results, and conclusions of a case study as a guidance for future design. We aim to answer the following research questions:

RQ1: Did the emergency remote teaching affect the academic performance of the IE201 course students?

RQ2: Are there any differences in students' academic performance between those who attended IE201 in a traditional F2F classroom, those who had a disrupted semester by the pandemic (BL), and asynchronous and synchronous online teaching modes?

RQ3: In an emergency transition, which group/s of IE201 students will be more affected in terms of academic performance, and which teaching modalities would be selected for each subgroup of students?

To answer these research questions, the study investigates the potential impact of different instruction modes (BL, asynchronous (Async.), and synchronous (Sync.) online teaching) resulting from the pandemic on the academic performance of undergraduate students enrolled in the IE201 course. The results are also compared with the traditional F2F classroom.

Realizing the effects of such closures on the academic performance of IE201 students is considered important for university-level planning and decision-making. The results might have some merits for other courses and instructors. The template model can be transferred to other courses, and employed by the university administrators, specifically for developing policies in emergency circumstances that are not limited to pandemics. The remainder of the paper is organized as follows: Section 2 details the materials and methods used in the analysis. The presentation of the data analysis and results are described in Section 3. Section 4 discusses the main research findings and limitations of the study, followed by conclusions in Section 5.

## 2. Materials and Methods

### 2.1. Research Methodology

As Alam and Parvin mentioned in their study, the effectiveness of an active learning process is often measured by its contribution to graduates' development [32]. Consequently, for primary and secondary education, academic performance is considered as the main parameter or key performance indicator (KPI) [32]. In addition to academic performance, the other key indicators for measuring the efficacy of an active learning process of higher education are also job-ready graduates, and the production of knowledge [30,34,35].

The aim of this project is to investigate how the emergency transition (from traditional F2F classrooms to online teaching modes) affected the academic performance of undergraduate students in the Financial Engineering course. Students are categorized into four cohorts based on the type of teaching modalities they attended. One cohort comprises pre-COVID-19 students that attended the IE201 course in the traditional face-to-face classroom. The second cohort consists of the students who attended the course in the transitional semester disrupted by the COVID-19 pandemic. The other two cohorts comprise during-COVID-19 cohorts. One of them is the students who attended the course in an asynchronous online teaching mode, and the other is those who attended the course in a synchronous online teaching mode.

We use students' IE201 course grades as the measure of academic performance, and transform it to a new relative metric called "Rank Percentage". The rank percentage is less sensitive to the absolute values of students' grades. We will discuss this evaluation metric in Section 2.2.

Given the data that we have, and the research time framework, the job-ready graduates and production of knowledge comparisons before and during the COVID-19 pandemic are not feasible. The reason is that they require standard questionnaires and longer time series data. It is worth mentioning that our comparison method is not an ideal one. Moreover, obtaining the information is challenging, and we are not trying to find the impact of the transition on every single student. Therefore, we indirectly address the problem by incorporating students' cumulative GPA (before the course started), and investigate the academic performance for the same sub-groups of students based on cumulative GPA bins. Four different cumulative GPA bins/subgroups are defined, based on letter grades of A, B, C, and D/F. We will discuss it in Section 2.4.

This research compares the rank percentage (a transformed version of the course grade) achieved by four cohorts of students in one domain, namely academic performance. Given the nature of the data, this study uses both descriptive analysis and statistical hypothesis tests to draw a more conclusive result. Therefore, based on Creswell's schema [36], herein, the research methodology is a mix of qualitative and quantitative, and the research model/design is a case study.

### 2.2. Rank Percentage Concept

In this study, we aim to investigate the effectiveness of different teaching modalities of the IE201 course before and during the COVID-19 pandemic. Educational researchers usually utilize course grades to evaluate students' academic performance [37–40]. The course grade has been considered as the performance measure of an individual student (absolute measure), and it is not comparable with other cohorts' grades. The delivery mode and education atmosphere were completely different for all four cohorts of students. So, under such circumstances, we propose a new metric called "Rank Percentage", and compare students' course rank percentage instead of students' course grades.

A rank is an ordinal number assigned to each student based on their performance in the class. In other words, after the final exam, students are sorted based on their final grades, and the rank will be assigned to each of them. Lesser rank means better performance and vice versa. Besides, since the course presented in each semester may differ in size, and to make comparison possible, the rank percentage is calculated using each student's rank divided by the total number of students in the class.

The rank percentage can capture students' academic performance compared to other students in the class. In addition, the rank percentage is not sensitive to the delivery mode and difficulty level of the course, and is always comparable for all students in the class. Moreover, in an emergency circumstance, such as the ERT resulting from the pandemic, considering students' grades as the evaluation metric to compare the effectiveness of different instructional modalities seems to be unreasonable.

*2.3. Course Data*

The study uses sample data from an undergraduate course called Financial Engineering (IE201), presented by the Mechanical and Industrial Engineering Department at the University of Illinois at Chicago. This is a theoretical and sophomore course taken by four different majors of Industrial, Mechanical, Civil Engineering, and Engineering Management, and needs intermediate calculus as the prerequisite. The data source contains course-level aggregated grades of students during the three affected semesters by the COVID-19 pandemic (Spring 2020, Fall 2020, and Spring 2021), and one previous unaffected semester (Spring 2019). They are the four cohorts of students previously introduced in Section 2.1. The general characteristics of the course are described in Table 1.

**Table 1.** IE201 general characteristics of four different instruction modes.

| Mode/Contribution to Final Grade [1] | Semester | Sample Size | #Exam [2] | #Homework [3] | Project | EXPO | COVID Affected? |
|---|---|---|---|---|---|---|---|
| F2F | Spring 2019 | 134 | 5<br>20% each | 11<br>5% | √<br>10% | √<br>5% | No |
| BL | Spring 2020 | 102 | 5<br>20% each | 11<br>10% | √<br>10% | - | Yes |
| Online (Async.) | Fall 2020 | 144 | 9<br>70% | 11<br>20% | √<br>10% | - | Yes |
| Online (Sync.) | Spring 2021 | 123 | 5<br>68% | 11<br>32% | - | - | Yes |

[1] In all four semesters (student cohorts), the lowest exam grade and the lowest homework grade for each student were dropped. [2] Number of exams. [3] Number of homework.

Spring 2019 (pre-COVID19 cohort) is considered as the traditional F2F instruction mode in which all classes were held in-person, whereas Spring 2020 (during COVID-19 cohort) is the transitional semester disrupted by the pandemic. So, almost half of the course was held in-person, and the remaining sessions were taught remotely using Blackboard and Zoom platforms. On the other hand, Fall 2020 and Spring 2021 were both completely online (during-COVID19 cohorts). The former was asynchronous using Blackboard Collaborate, and the latter was synchronous using the Zoom platform and Blackboard. In addition, since the IE201 withdrawal rate is very low, students who withdrew from the course are not included in our analysis. Furthermore, the course was taught by the same instructor in all semesters.

*2.4. Student Data*

Cumulative grade point average (GPA) is considered an important indicator of the academic history of students. So, we incorporate students' cumulative GPA (before the IE201 course started) to find out the academic history of students in each cohort. Figure 1 illustrates the distribution of cumulative GPAs achieved by students enrolled in the IE201 course (before the course started) in all four semesters.

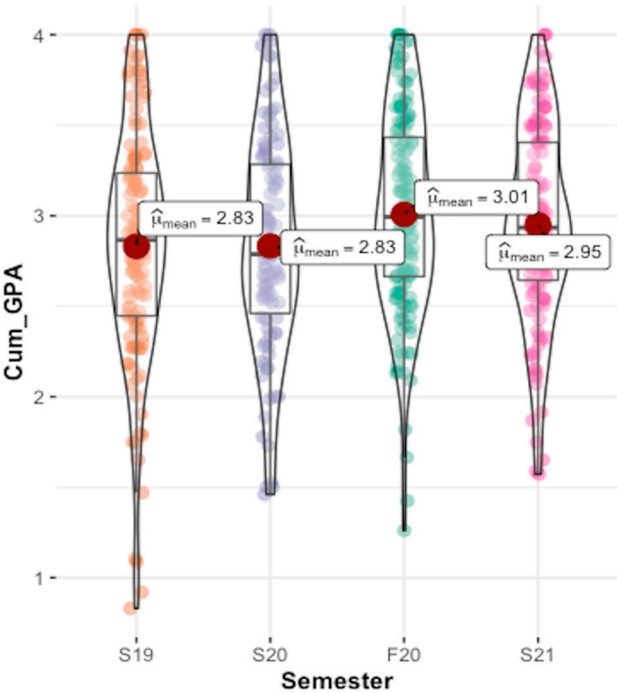

**Figure 1.** Differences in the distribution of cumulative GPA in the semesters.

As the graphs present, if we disregard a few outliers, the dispersion of the cumulative GPAs is almost the same for all cohorts. So, all cohorts of students come from almost similar academic history.

Figure 2 represents the scatterplots of cumulative GPA versus rank percentage for all four cohorts. We employ the "Locally Estimated Scatterplot Smoothing" (LOESS) method for fitting a smooth curve between two variables of cumulative GPA and rank percentage. A span value of 0.35 is also utilized to control the degree of smoothness. As the graph suggests, in all instruction modalities, the rank percentage values decrease as the cumulative GPAs are increasing, i.e., it can be observed that there exists a potential inverse relationship between the cumulative GPA and rank percentage.

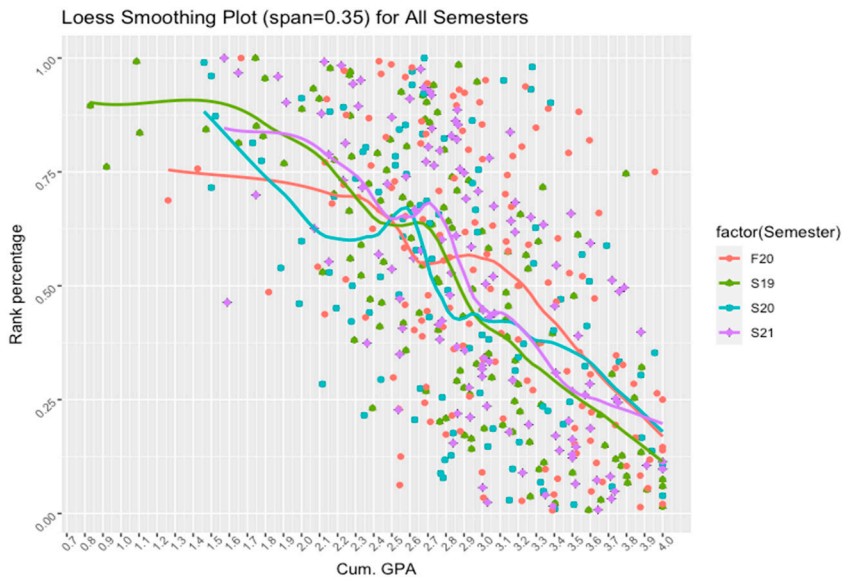

**Figure 2.** Cumulative GPA vs. rank percentage in four instruction modes.

On the other hand, comparing the curves seems to be not conclusive regarding the students' academic performance. This means, depending on the cumulative GPA spectrum, the relative rank percentage of the curves is changing with respect to each other. Moreover, decisions based on the 0.00 to 4.00 cumulative GPA boundary could be controversial. In other words, there might be some hidden trends that can be detected if we divide each cohort of students into certain subgroups based on a specific criterion, such as cumulative GPA.

Therefore, to measure the potential impacts of the ERT on students' academic performance in IE201, we consider cumulative GPA bins. To define cumulative GPA bins or subgroups, we employ the norm that the majority of the engineering faculty used to determine letter grades. The subgroups include G1 = (3.40, 4.00), G2 = (2.90, 3.40), G3 = (2.40, 2.90), and G4 = (0, 2.40) which stand for A, B, C, and D/F letter grades, respectively. Therefore, we compare the academic performance of each subgroup (rank percentage) between different cohorts. It seems that there could be a considerable difference between students' rank percentage in each semester when the cumulative GPA subgroups changed.

*2.5. Statistical Hypothesis Tests*

As we described, the rank percentage is considered to evaluate students' academic performance in each cohort, and we divided each cohort of students based on their cumulative GPAs into four different subgroups. The goal is to compare students' rank percentage in each cumulative GPA subgroup between the four various cohorts (teaching modalities) to test if there exist any significant differences.

To analyze the data and present them in the findings, firstly, we use descriptive analysis using some graphs (violin plots) and simple statistical parameters such as mean. Secondly, statistical methods are also used to draw more conclusive results. The tests include the Kruskal–Wallis test for differences in academic performance across all four cohorts, and Mann–Whitney U tests to test for differences in rank percentage between each pair of the cohorts. In Mann–Whitney and Kruskal–Wallis tests, the test statistic only depends on the ranks of the observations, and no assumption about the distribution of the population is made. The former is used for two samples, whereas the latter is used when there are two or more samples. These non-parametric tests are employed because our observations do not follow the normality assumption. Statistical tests are also performed using R software (version 4.0.3) and the R package "ggstatsplot", with the most common analysis options combined with a graphical output [41].

**3. Results**

We investigate the potential impact of different teaching modalities caused by the ERT on the academic performance of IE201 students. Specifically, we implement the following three scenarios to study which teaching modalities would be more effective for each subgroup of IE201 students in an emergency transition to online modes:

Scenario 1: Comparing students' rank percentage over the four cohorts of students who attended different instructional modalities, including F2F, BL, Async., and Sync.

Scenario 2: Comparing students' rank percentage between F2F mode (the pre-COVID-19 cohort) and all three semesters affected by the COVID-19 pandemic (BL, Async., and Sync., or during-COVID-19 cohorts).

Scenario 3: Comparing students' rank percentage between F2F mode and online modes (Async. and Sync.).

Further, to clarify the differences between instructional modalities that we consider in this study; we again describe them here:

F2F (S19): Traditional face-to-face classroom, and not affected by the pandemic (pre-COVID-19 cohort).

BL (S20): Transitional semester disrupted by the COVID-19.

Async. (F20): Asynchronous online instruction mode.

Sync. (S21): Synchronous online instruction mode.

It is worth mentioning that the last three teaching modes were affected by the COVID-19 pandemic.

### 3.1. Scenario 1: Comparing F2F, BL, Async., and Sync.

Figure 3 depicts the distribution of students' rank percentage for each of the cumulative GPA subgroups in all four cohorts of students (i.e., different teaching modes). The graphs suggest that there are some differences between the teaching modes' effectiveness in some subgroups, such as subgroup G1 (top left graph) and G4 (bottom right graph). So, we investigate them using the Kruskal–Wallis test.

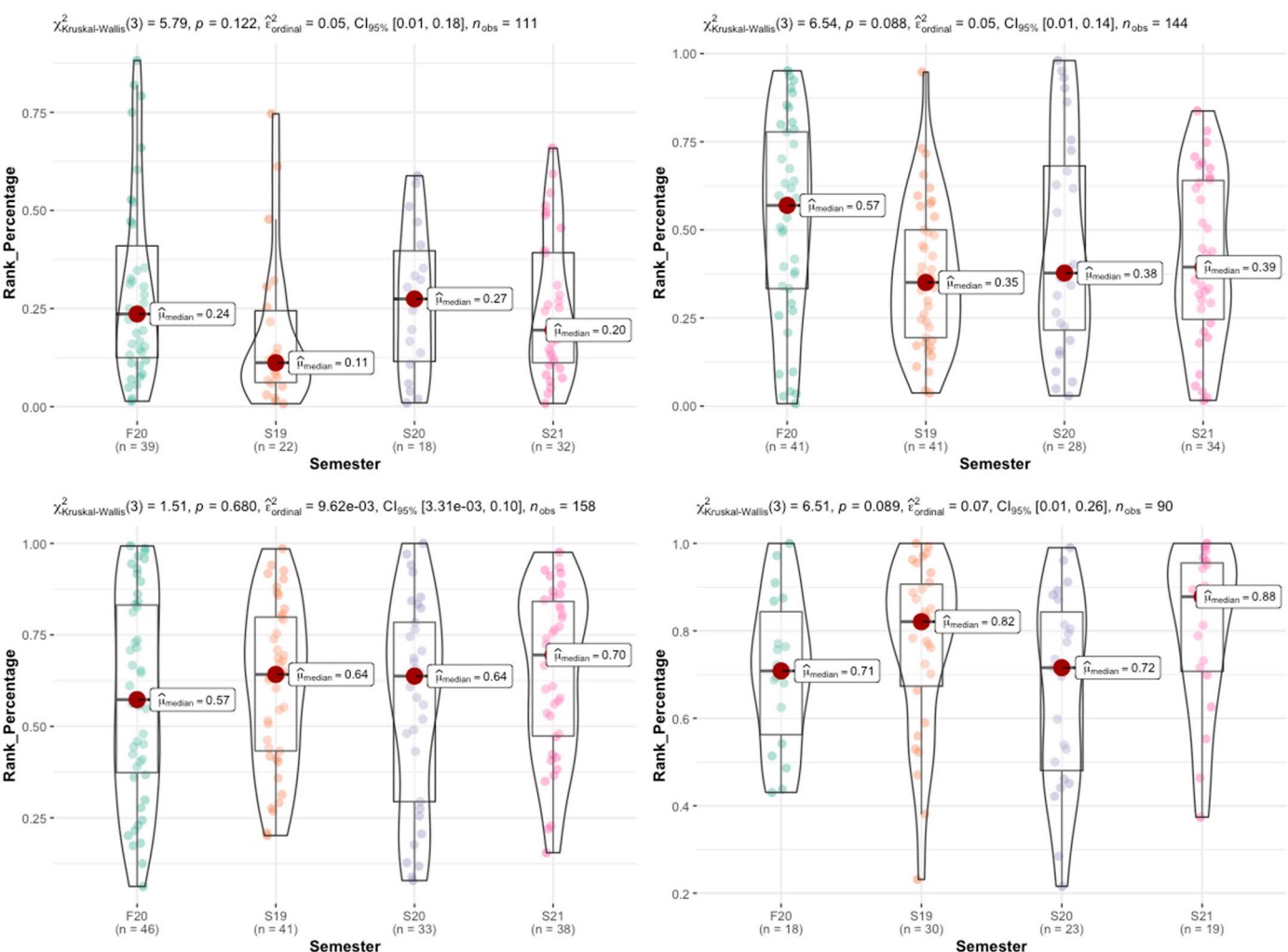

**Figure 3.** Differences in the distribution of rank percentage between four various instruction modes in cumulative GPA subgroup G1 (**top left**), subgroup G2 (**top right**), subgroup G3 (**bottom left**), and subgroup G4 (**bottom right**).

In Table 2, we represent the result of statistical tests to check the differences in students' rank percentage throughout all four instructional modalities.

**Table 2.** *P*-value of Kruskal–Wallis test comparing all modes.

| Terms/Cum. GPA Subgroups | G1 | G2 | G3 | G4 |
|---|---|---|---|---|
| S19, S20, F20, S21 | $1.223 \times 10^{-1}$ | $8.801 \times 10^{-2}$ | $6.797 \times 10^{-1}$ | $8.926 \times 10^{-2}$ |

As the results show, there are no significant differences between all modes, i.e., at the significant level of 5 percent, and when we compare all teaching modes together (the academic performance of four cohorts of students in each cumulative GPA subgroup), none of them are significant. So, none of the teaching modes are more effective than the others. On the other hand, when we compare the output of the graphical tests illustrated in Figure 3, it seems that for some cohorts, the statistical test results can be different if we employ pairwise comparisons (Mann–Whitney U test).

Table 3 summarizes pairwise comparison test results between different instructional modalities in IE201. Some of the differences are statistically significant. In these cases, we also employ the one-tailed hypothesis to investigate in which modes (cohorts) students performed better. In cumulative GPA subgroup G1, there is a significant difference between F2F and Async. modes. As the one-tailed test result shows in subgroup G1, the academic performance of students in the asynchronous online mode is worse than those in traditional F2F class. This difference is even more significant when we compare F2F and Async. modes in subgroup G2. This suggests that the transition from face-to-face to an asynchronous online mode affected academically average-to-good students more. Moreover, in subgroup G1, students' academic performance in the F2F mode is better than the online synchronous mode (one-tailed *p*-value = 0.0369).

**Table 3.** *p*-value of Mann–Whitney tests comparing all modes.

| Terms/Cum. GPA Sub. | G1 (2-Tailed) | G1 (1-Tailed) | G2 (2-Tailed) | G2 (1-Tailed) | G3 (2-Tailed) | G3 (1-Tailed) | G4 (2-Tailed) | G4 (1-Tailed) |
|---|---|---|---|---|---|---|---|---|
| **S19, S20** | $1.255 \times 10^{-1}$ | - | $4.19 \times 10^{-1}$ | - | $5.962 \times 10^{-1}$ | - | $7.7 \times 10^{-2}$ | $3.85 \times 10^{-2}$ * H1:S19 > S20 |
| **S19, F20** | $2.029 \times 10^{-2}$ * | $1.014 \times 10^{-2}$ * H1:F20 > S19 | $9.037 \times 10^{-3}$ ** | $4.519 \times 10^{-3}$ ** H1:F20 > S19 | $6.326 \times 10^{-1}$ | - | $1.729 \times 10^{-1}$ | - |
| **S19, S21** | $7.379 \times 10^{-2}$ | $3.69 \times 10^{-2}$ * H1:S21 > S19 | $3.763 \times 10^{-1}$ | - | $5.036 \times 10^{-1}$ | - | $5.114 \times 10^{-1}$ | - |
| **S20, F20** | $8.502 \times 10^{-1}$ | - | $3.275 \times 10^{-1}$ | - | $9.208 \times 10^{-1}$ | - | $7.651 \times 10^{-1}$ | - |
| **S20, S21** | $7.564 \times 10^{-1}$ | - | $7.734 \times 10^{-1}$ | - | $3.042 \times 10^{-1}$ | - | $3.495 \times 10^{-2}$ * | $1.748 \times 10^{-2}$ * H1:S21 > S20 |
| **F20, S21** | $4.868 \times 10^{-1}$ | - | $9902 \times 10^{-2}$ | $4.951 \times 10^{-2}$ * H1:F20 > S21 | $3.229 \times 10^{-1}$ | - | $8.88 \times 10^{-1}$ | $4.44 \times 10^{-2}$ * H1:S21 > F20 |

Note: * indicates statistical significance at the 5% level. ** indicates statistical significance at the 1% level.

In addition, in subgroup G4, or students with cumulative GPAs below 2.40, there are some significant differences between F2F and BL, BL and Sync., and Async. and Sync. modes. The results suggest that this subgroup of students performed better in the blended transitional semester disrupted by the pandemic in comparison with face-to-face and online synchronous modes.

On the other hand, in subgroup G3, high *p*-values indicate that the evidence is not strong enough to suggest an effect in the population. In other words, the equality assumption of the medians of students' rank percentage in all instruction modes cannot be rejected. So, we can assume that the ERT has no considerable effect on the performance of this group of students.

### 3.2. Scenario 2: F2F vs. BL-Async.-Sync. (Pre-COVID-19 vs. during-COVID-19 Cohorts)

Table 4 concludes the results of comparing the academic performance between F2F mode and all three COVID-19-affected semesters to investigate the potential impacts of ERT. In subgroups G1 and G2, at the level of 5 percent, the differences are significant. As the one-tailed hypothesis suggests, students with cumulative GPAs greater than 2.90 have been more affected by the emergency transition remote teaching, and their academic performance has been negatively impacted. In other words, in subgroups G1 and G2, the pre-COVID-19 cohort achieved better academic grades compared to during-COVID-19 counterparts.

**Table 4.** *p*-value of Mann–Whitney tests comparing F2F vs. BL-online modes.

| Terms/Cum. GPA Sub. | G1 (2-Tailed) | G1 (1-Tailed) | G2 (2-Tailed) | G2 (1-Tailed) | G3 (2-Tailed) | G3 (1-Tailed) | G4 (2-Tailed) | G4 (1-Tailed) |
|---|---|---|---|---|---|---|---|---|
| S19 vs. S20-F20-S21 | $2.476 \times 10^{-2}$ * | $1.238 \times 10^{-2}$ * H1:S20-F20-S21 > S19 | $5.47 \times 10^{-2}$ | $2.735 \times 10^{-2}$ * H1:S20-F20-S21 > S19 | $7.631 \times 10^{-1}$ | - | $3.292 \times 10^{-1}$ | - |

Note: * indicates statistical significance at the 5% level.

### 3.3. Scenario 3: F2F vs. Online

Combining asynchronous and synchronous online modes to compare them with the traditional F2F class is considered in this scenario. Table 5 confirms the same, yet more significant, results compared to the second scenario. It suggests that students with A and B letters' cumulative GPA grades performed better in terms of academic scores in the traditional F2F class compared to online teaching.

**Table 5.** *p*-value of Mann–Whitney tests comparing F2F vs. online modes.

| Terms/Cum. GPA Sub. | G1 (2-Tailed) | G1 (1-Tailed) | G2 (2-Tailed) | G2 (1-Tailed) | G3 (2-Tailed) | G3 (1-Tailed) | G4 (2-Tailed) | G4 (1-Tailed) |
|---|---|---|---|---|---|---|---|---|
| S19 vs. F20-S21 | $1.992 \times 10^{-2}$ * | $9.959 \times 10^{-3}$ ** H1:F20-S21 > S19 | $2.618 \times 10^{-2}$ * | $1.309 \times 10^{-2}$ * H1:F20-S21 > S19 | $9.685 \times 10^{-1}$ | - | $6.59 \times 10^{-1}$ | - |

Note: * indicates statistical significance at the 5% level. ** indicates statistical significance at the 1% level.

## 4. Discussion

This section describes responses to the research questions that we raised in the introduction section. The first question is about the impact of the ERT on the IE201 course students in terms of academic performance. Generally speaking, the analysis reveals that there is no significant difference between students' academic performance when we compare all four cohorts of students. However, pairwise comparisons reveal that specific subgroups of students have been affected by the emergency transition to remote teaching. It is worth keeping in mind that the analysis of the transitional semester disrupted by the COVID-19 pandemic (Spring 2020: BL) is considered as a report, and we cannot draw any strong conclusion based on its results.

Regarding the second research question (i.e., differences in students' academic performance), three different comparisons can be considered: (1) F2F vs. Asynchronous: The results suggest that differences are significant in cumulative GPA subgroups G1 and G2. In other words, students with cumulative GPAs greater than 2.90, who are considered academically average-to-good students, have performed worse in asynchronous online teaching compared to the traditional face-to-face classroom. It seems that the academic performance of students in other cumulative GPA subgroups (G3 and G4) was not statistically different. (2) F2F vs. Synchronous: The analysis supports that students with cumulative GPAs above 3.40 (subgroup G1) performed better in the face-to-face class in comparison with synchronous online instruction. It is also observed that there are not any statistically significant differences in other subgroups. It seems that the majority of students had a reasonable academic performance with the synchronous instruction mode. (3) Synchronous vs. Asynchronous: The differences are considered significant in two cumulative GPA subgroups: subgroups G2 and G4. The results reveal that subgroup G2 students performed better in the synchronous online instruction compared to the asynchronous one, whereas the academic performance of students with cumulative GPAs below 2.40 was better in the asynchronous online mode compared to the synchronous one. The results also suggest that the difference between asynchronous and synchronous online instruction modes is not very conclusive.

The third question concerns the effectiveness of different teaching modalities in an emergency transition. The analysis, particularly the results of second and third scenarios, would support that among all subgroups of students in different cohorts, the academic performance of students with cumulative GPAs above 2.90 (subgroups G1 and G2) have been

negatively impacted by the transition to online education. In other words, in subgroups G1 and G2, the pre-COVID-19 cohort achieved better academic grades in comparison with the during-COVID-19 cohorts. It seems that these subgroups of students are more dependent on the face-to-face classroom. So, for further decisions regarding the instructional modality design, this consideration could be taken into account. For instance, non-mandatory small-sized classes could be implemented for these groups of students. On the other hand, it seems that students with cumulative GPAs below 2.90 have been not significantly affected by the transition to online modes. They could be more flexible in terms of instructional modality design.

It is worth noting that Russell's book lists 355 sources dating back as early as 1928 to discuss compelling arguments, and settle the debate of online learning and its effectiveness, specifically in comparison to face-to-face learning [42]. The general conclusion of the evidence renders is that there is no significant difference to be almost indisputable. Russell notes that just because the research suggests that there is no difference in student performance, this does not mean that distance learning is necessarily better than other methods of learning, just that it can be as effective. However, there are also some criticisms about Russell's work, such as that it failed to control for extraneous variables or use valid tools to measure outcome [18,43]. In the current research, Russell's conclusion is supported if we compare all four cohorts (four different teaching modalities), i.e., no significant difference exists. However, when we split the cohorts to the subgroups based on criteria (cumulative GPA here), there are some significant differences. This also might be related to the fact that there are some extraneous variables, and the cumulative GPA could be considered as one of them.

The results of this study represent that the COVID-19 pandemic has affected the academic performance of undergraduate students who attended the Financial Engineering course. Although previous studies mainly focus on students' course grades as an academic performance evaluation metric, we define the rank percentage measure to test our hypothesis, which is unique to the literature. We do not claim that the rank percentage is the best metric. Since the delivery mode and education atmosphere were completely different for all cohorts, there is not an ideal metric to compare the effectiveness of different teaching modes. However, the rank percentage allows us to make the academic scores more comparable. There is a wide range of factors that might affect students' academic performance, such as classroom population, academic history of the instructor, level of the class, major, university entrance score, etc. [37]. Some of these factors are controlled in our analysis. For instance, since the same instructor taught the course for all of the cohorts, the impact of the academic history of the instructor is controlled. The difficulty level of the classes is also controlled by introducing the rank percentage metric. Moreover, the academic level of students attended in the class is reflected in the cumulative GPAs, which has been used to define more academically homogenous subgroups for our comparison purposes.

Moreover, in this study, we only consider one research domain, namely academic performance. Given the data and research time frame, multidimensional analysis is not considered. We attempt to reveal that although the differences between all cohorts are not significant, certain subgroups of students have been impacted by the transition. We investigate this assumption by splitting each cohort of students into subgroups based on specific criterion, such as cumulative GPA. For instance, in our study, academically average-to-good students with cumulative GPAs greater than 2.90 have been negatively impacted. This impact is not detectable without dividing the cohorts into specific subgroups. Defining meaningful subgroups and splitting criteria can be the subjects for future research. The reason for choosing cumulative GPA (splitting criterion) is to create subgroups that have at least one similar interpretable attribute.

This study has certain limitations. Firstly, it is considered a case study in the Financial Engineering course, and the results are specific to one course taught by a single instructor in one higher education institution. In addition, the choice of the course was made by convenience and availability of data. So, we do not claim the universal validity of our

findings, and for organizational-level decisions, more studies need to be done. Moreover, there are some insignificant differences (in subgroup G3, for instance) that might represent some hidden patterns, and cannot be detected with this research design.

Secondly, some individual aspects, such as students' digital skills, the accessibility and ownership of digital technologies, and self-regulation, are not considered in this study, and can be used to develop a questionnaire for a future study.

Thirdly, there is enough evidence that during the pandemic, students have experienced a lot of stress [44]. The stress caused by the pandemic lockdowns is closely related to anxiety, loneliness, and depression [45]. Therefore, it could have a potential impact on students' academic performance decrease. Since the attitudes and behavior of students have been not considered in this study, the potential negative effects of such variables are not evaluated. This limitation also includes other COVID-19-related issues, such as medical and financial problems.

Finally, transparency, reliability, and security issues of online evaluation and examination have always been controversial [46]. So, the potential effect of cheating behavior cannot be discarded. We could not assess dishonest behavior in our analysis.

Considering the sample sizes, this study does not represent a general comparison model for the teaching modalities. We suggest that our findings invite the research community to seek or investigate the effectiveness of different teaching modes in terms of academic performance on subgroups of students when comparing populations with different teaching modes. Though our work has its limitations, it certainly encourages the readers to navigate this line of research, and focus their studies on certain subgroups of interest.

## 5. Conclusions

The unprecedented global health crisis has prompted emergency adaptations to a distance teaching-learning system called "emergency remote teaching" (ERT). There are a lot of concerns about the effectiveness of the shift to online learning among students, faculty, and higher education administrators. This study is an effort to investigate the potential impacts of such a transition on the academic performance of students enrolled in the Financial Engineering course. We have employed a novel rank percentage measure to compare students' academic performance in a transitional disrupted semester by the pandemic, two consecutive online semesters, and a traditional face-to-face classroom. Our analysis reveals that the differences are significant between specific subgroups of students. The findings suggest that the academic performance of students with cumulative GPAs greater than 2.90, specifically higher than 3.40, has been negatively impacted by the transition, whereas the impact on students with cumulative GPAs below 2.90 are not very conclusive.

The COVID-19 pandemic should be considered as an opportunity to enhance digital preparedness, capacity development, and innovations in higher education institutions. This study aims to assist university administrators to make decisions about short or long-term closures, re-opening face-to-face classes, and online learning continuance in extreme situations, disruptions, and emergency circumstances.

**Author Contributions:** Conceptualization, R.N. and H.D.; methodology, R.N. and H.D.; validation, R.N., H.D. and P.C.N.; formal analysis, R.N. and H.D.; resources, R.N., H.D. and P.C.N.; data curation, R.N., H.D. and P.C.N.; writing—original draft preparation, R.N.; writing—review and editing, R.N., H.D. and P.C.N.; visualization, R.N.; supervision, H.D. All authors have read and agreed to the published version of the manuscript.

**Funding:** This research received no external funding.

**Institutional Review Board Statement:** This study was approved by University of Illinois at Chicago Internal Review Board. Permission from University of Illinois at Chicago Privacy Board and Internal Review Board were required to access the data used in this study. All the experiment protocols involving human data were in accordance with the University of Illinois at Chicago Privacy Board and Internal Review Board guidelines.

**Informed Consent Statement:** Our research was provided a waiver of informed consent, parental permission and assent from the University of Illinois at Chicago IRB granted under 45 CFR 46.116(f).

**Data Availability Statement:** The data presented in this study are available on request from the corresponding author. The data are not publicly available due to privacy concerns.

**Acknowledgments:** The authors acknowledge the help of James Muench in preparing and de-identifying the data used in this research.

**Conflicts of Interest:** The authors declare no conflict of interest.

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
