# Peer review of "Impacts on Students’ Academic Performance Due to Emergency Transition to Remote Teaching during the COVID-19 Pandemic: A Financial Engineering Course Case Study"

_education, doi:10.3390/educsci12030202_

Round 1

Reviewer 1 Report

  • In the following expression, I guess that authors refer to blended learning. “Synchronous and asynchronous are two types of on-line instruction mode when considering synchrony, while a mixed version of both modes is also possible”. Please see the following reference: In pursuit of the right mix: Blended learning for augmenting, enhancing, and enriching flexibility. Asian Journal of Distance Education, 16(2), i-vi. https://doi.org/10.5281/zenodo.5827159
  • Authors can refer to following article in the literature review section where they report review studies: Research trends in online distance learning during the COVID-19 pandemic. Distance Education, 1-26. https://doi.org/10.1080/01587919.2021.1986373
  • At the beginning of the methodology section “2.1. Research Methodology”, the authors can clearly state the methodology (qual, quan, or mixed) and research model/design (e.g., survey, case study, phenomenology, experimental study, explanatory sequential mixed design, etc.). Authors can adopt Creswell’s schema. It is obvious that the research adopts quantitative paradigm, but this should be explicitly articulated.
  • Creswell, J. W. (2004). Educational Research: Planning, Conducting, and Evaluating Quantitative and Qualitative Research. Pearson.
  • Authors can benefit from the following reference in the discussion section. It would be good to report counter arguments: T. L. Russel , The no significant difference phenomenon as reported in 355 research reports, summaries and papers. Raleigh, NC: North Carolina State University, 1999.
  • I suggest improving suggestions/implications section.

Reviewer 2 Report

Dear Author, Thank you so much. I have read your contribution which is contemporary and well-timed. However, this needs a number of development before it may reach to a publishable standard. I provide some comments and guideline for further improvement.

Firstly, introduction is too long to have focus. Many information that is readily available on other sources and do not need to repeat in this study.  It is rather important to explore the research problem and gap by introducing latest works and practical examples. You have missed to explore the problem and scoping the research although you have used a great amount of current literature.   

Secondly, in research method, you need to establish learning domains to collect data and to analyse them. The learning domains must include both theoretical and practical domains. This section is very weak. Please visit following paper to understand how learning domains can be set the courses of business education. 

  • Does MBA’s paradigm transformation follow business education’s philosophy? A comparison of academic and job-performance and SES among five types of MBAian     
  • Does an MBA degree advance business management skill or in fact create horizontal and vertical mismatches?    

You have used rank percentage idea which is good but you need to explain why this idea is valid in eye of statistical analysis which is very important. You may have further idea on it from “Can online higher education be an active agent for change? —comparison of academic success and job-readiness before and during COVID-19,”; Online technology: sustainable higher education or diploma disease for emerging society during emergency—comparison between pre and during COVID-19 and Does online technology provide sustainable HE or aggravate diploma disease? Evidence from Bangladesh—a comparison of conditions before and during COVID-19, which you already used in your paper as references.    

*** Without setting domains and making comparison of them, learning assessment can NOT be made. Please make sure it is done.

Thirdly, please avoid using bullet points in academic writing. In presenting results, you have used bullet points, please avoid it and please explain the results with the learning assessment domains as suggested.

Fourthly, in the discussions you have written question form, please make an assertive form of them.

Sixthly, both theoretical and practical implications along with limitations are not impressive. Please incorporate them.

Finally, a serious effort is needed to edit the texts.

While, I have suggested a number of corrections, they are addressable with a considerable amount of attention. I shall be happy to read the revised version to see if the development is made.

Good luck with revision.     

Round 2

Reviewer 2 Report

The necessary improvements are made. However an English editing may help.